# The Intersection of Epigenetics and Senolytics in Mechanisms of Aging and Therapeutic Approaches

**DOI:** 10.3390/biom15010018

**Published:** 2024-12-26

**Authors:** Daiana Burdusel, Thorsten R. Doeppner, Roxana Surugiu, Dirk M. Hermann, Denissa Greta Olaru, Aurel Popa-Wagner

**Affiliations:** 1Experimental Research Center for Normal and Pathological Aging, University of Medicine and Pharmacy Craiova, 200349 Craiova, Romania; daiana.burdusel@gmail.com (D.B.); roxana.surugiu07@gmail.com (R.S.); dirk.hermann@uk-essen.de (D.M.H.); 2Department of Neurology, University of Giessen Medical School, 35392 Giessen, Germany; thorsten.doeppner@neuro.med.uni-giessen.de; 3Department of Neurology, University Medical Center Göttingen, 37075 Göttingen, Germany; 4Chair of Vascular Neurology and Dementia, Department of Neurology, University Hospital Essen, 45147 Essen, Germany

**Keywords:** epigenetics, senolytics, cellular senescence, aging mechanisms, epigenetic clock, DNA methylation, healthspan extension

## Abstract

The biological process of aging is influenced by a complex interplay of genetic, environmental, and epigenetic factors. Recent advancements in the fields of epigenetics and senolytics offer promising avenues for understanding and addressing age-related diseases. Epigenetics refers to heritable changes in gene expression without altering the DNA sequence, with mechanisms like DNA methylation, histone modification, and non-coding RNA regulation playing critical roles in aging. Senolytics, a class of drugs targeting and eliminating senescent cells, address the accumulation of dysfunctional cells that contribute to tissue degradation and chronic inflammation through the senescence-associated secretory phenotype. This scoping review examines the intersection of epigenetic mechanisms and senolytic therapies in aging, focusing on their combined potential for therapeutic interventions. Senescent cells display distinct epigenetic signatures, such as DNA hypermethylation and histone modifications, which can be targeted to enhance senolytic efficacy. Epigenetic reprogramming strategies, such as induced pluripotent stem cells, may further complement senolytics by rejuvenating aged cells. Integrating epigenetic modulation with senolytic therapy offers a dual approach to improving healthspan and mitigating age-related pathologies. This narrative review underscores the need for continued research into the molecular mechanisms underlying these interactions and suggests future directions for therapeutic development, including clinical trials, biomarker discovery, and combination therapies that synergistically target aging processes.

## 1. Introduction

The study of aging has evolved significantly, with recent research bringing epigenetics and senolytics into the spotlight as important approaches to understanding the biological mechanisms underlying aging and age-related diseases. Aging is a multifaceted biological process characterized by a gradual decline in physiological functions and an increased susceptibility to various diseases, including cancer, cardiovascular disorders, and neurodegenerative conditions. The interaction between genetic predispositions and environmental influences, such as diet, lifestyle, and stress, contributes to the complexity of aging [1,2,3]. Epigenetics refers to heritable changes in gene expression that occur without alterations to the DNA sequence itself. These changes can influence cellular behavior and function throughout an organism’s life. Recent studies have established that epigenetic modifications, such as DNA methylation and histone modifications, regulate gene expression patterns contributing to cellular senescence and tissue dysfunction [4,5]. For example, epigenetic clocks, like the Horvath clock, measure biological age by assessing DNA methylation patterns and can predict an individual’s risk of age-related diseases [5]. Targeting senescent cells through therapies that clear these cells may reverse or slow age-related deterioration [6]. Techniques in epigenetic reprogramming, such as using Yamanaka factors, show promise in partially reversing biological age in cells, aiding in tissue regeneration [4,7]. Additionally, epigenetic biomarkers facilitate personalized aging interventions, allowing tailored lifestyle or dietary adjustments to reduce biological aging markers and improve healthspan [8]. These applications underscore how epigenetics advances aging research and suggests new therapeutic strategies to counteracted its effects. On the other hand, senolytics are a class of compounds specifically designed to target and facilitate the removal of senescent cells, which have undergone irreversible growth arrest but remain metabolically active and can secrete inflammatory factors that contribute to tissue dysfunction [9]. The accumulation of senescent cells is linked to various age-related pathologies, including cardiovascular disease, osteoarthritis, pulmonary fibrosis, and neurodegenerative disorders [10]. These cells secrete inflammatory molecules, proteases, and other factors collectively known as the senescence-associated secretory phenotype (SASP), contributing to tissue damage and systemic inflammation and accelerating aging processes [11,12]. For example, in cardiovascular disease, senescent cells in vascular tissue promote inflammation and stiffness, factors associated with hypertension and atherosclerosis [13]. In osteoarthritis, senescent cells contribute to cartilage degradation and joint inflammation [14], while in pulmonary fibrosis, senescent cells worsen fibrosis, impairing lung function [15]. Evidence also suggests that brain cell senescence contributes to neuroinflammation, linked to Alzheimer’s disease progression [16].

Cellular senescence is a state of stable cell cycle arrest that occurs in response to stressors such as DNA damage, telomere shortening, and oncogenic signals [17]. While senescence acts as a protective mechanism against tumorigenesis by preventing the proliferation of damaged cells, the accumulation of senescent cells over time disrupts tissue homeostasis, fosters chronic inflammation, and impairs the regenerative potential of adult stem cells by altering the stem cell niche, essential for tissue repair [18,19,20]. SASP factors recruit immune cells to sites of tissue damage, exacerbating inflammation and promoting secondary senescence in neighboring cells [21], creating a cycle where inflammation drives further tissue damage and accelerates the aging process. Senescent cells also resist apoptosis through the activation of survival pathways mediated by the BCL-2 family of proteins, allowing their persistence and contributing to their harmful effects [22]. For instance, senescent chondrocytes in osteoarthritis secrete MMPs, contributing to cartilage degradation, while senescent endothelial cells promote vascular aging and atherosclerosis [4,23,24]. This scoping review explores the intricate relationship between epigenetics and senolytics in the context of aging and their therapeutic potential.

## 2. Materials and Methods

The literature search for this scoping review was conducted across four major databases, PubMed, Web of Science, Google Scholar, and ClinicalTrials.gov, to ensure inclusive coverage of relevant studies. Following the Preferred Reporting Items for scoping review guidelines, we meticulously developed the search strategy to ensure transparency and reproducibility. Detailed search strings were constructed with specific keywords such as “DNA Methylation”, “Histone Modification”, “senolytic agents”, and “aging processes” to capture the full scope of literature on epigenetics, senolytics, and aging. PubMed’s Medical Subject Headings (MeSHs) were used to refine and optimize search terms related to epigenetic mechanisms, senolytic agents, and aging. The Web of Science platform included multidisciplinary research through citation indexing, while Google Scholar expanded access to non-indexed sources. We limited our search to English-language papers published within the past 20 years to maintain relevance and currency. The initial study selection was based on abstract screening by two independent reviewers, with any discrepancies resolved by consensus. This approach ensured a final set of studies reflecting a robust cross-section of the literature, incorporating diverse perspectives and research findings.

## 3. Epigenetics in Aging

Environmental factors influence aging by interacting with epigenetic mechanisms such as DNA methylation, histone modifications, chromatin remodeling, and non-coding RNAs (Figure 1). These mechanisms affect gene expression and contribute to cellular aging and the development of age-related diseases. Environmental stressors like diet, smoking, circadian rhythm disruption, psychological stress, and genetic predisposition have been associated with epigenetic changes that influence the aging process [4,25,26].

Diet can influence epigenetic modifications, particularly through the mechanisms of DNA methylation. Nutrients such as folate, methionine, and vitamins B6 and B12 are vital for maintaining proper DNA methylation processes. A high-fat diet (HFD) has been shown to induce global DNA hypomethylation, which can compromise genomic stability and elevate the risk of metabolic diseases, including diabetes and cardiovascular conditions [27,28,29].

Conversely, certain dietary patterns, such as caloric restriction and the Mediterranean diet, which is rich in antioxidants and anti-inflammatory foods, have been associated with beneficial epigenetic changes. These diets can activate SIRT1, a protein involved in DNA repair and longevity [30]. Furthermore, foods that are high in methyl donors, such as leafy greens, fish, and legumes, support DNA methylation processes. This support may offer protective effects against conditions like cognitive decline and osteoporosis [31,32].

Smoking is another environmental factor that induces epigenetic alterations, particularly in genes involved in inflammation, tumor suppression, and immune regulation. Smoking has been linked to hypermethylation of tumor suppressor genes, such as p16INK4a, which increases the risk of cancers, particularly lung cancer. Smokers also often display global DNA hypomethylation, contributing to genomic instability and increasing the likelihood of developing diseases such as lung cancer, chronic obstructive pulmonary disease and atherosclerosis [33,34,35]. Interestingly, some of the epigenetic damage resulting from smoking, particularly changes in DNA methylation, can be partially reversed after quitting smoking. Studies have shown that the recovery of DNA methylation levels occurs over time following cessation, suggesting potential for epigenetic recovery post-smoking [33,36,37]. For instance, a study found that many differentially methylated CpG sites returned to levels comparable to non-smokers within five years after quitting [33].

Disruption of the circadian rhythm, whether due to shift work, irregular sleep patterns, or jet lag, affects gene expression and cellular processes by inducing epigenetic changes [38,39]. Circadian genes, such as PER1 and BMAL1, regulate metabolism, cellular repair, and inflammatory responses [40,41,42]. When disrupted, these genes undergo epigenetic changes, particularly in DNA methylation patterns, which have been linked to metabolic disorders, immune system dysregulation, and neurodegenerative conditions [43]. Additionally, chronic circadian disruption can accelerate aging through mechanisms such as telomere shortening, further contributing to age-related diseases like cancer and cardiovascular disease. For instance, studies have shown that circadian disruption can lead to altered clock gene expression that adversely affects metabolic health outcomes, including obesity and type 2 diabetes [44]. Furthermore, the molecular mechanisms underlying these changes involve complex feedback loops among core clock genes that regulate a substantial portion of the human genome [45].

Psychological stress also contributes to aging through its impact on the hypothalamic–pituitary–adrenal (HPA) axis, which controls the body’s response to stress. Chronic stress can induce epigenetic modifications, especially through changes in DNA methylation of stress-response genes, such as FKBP5, leading to conditions such as cognitive decline, depression, and cardiovascular disorders. Stress-induced epigenetic changes also promote cellular senescence, accelerating biological aging and increasing vulnerability to diseases like Alzheimer’s disease and diabetes. These epigenetic changes can persist over time, particularly in individuals exposed to early-life stress, resulting in a higher predisposition to age-related diseases [46,47].

Genetic predispositions interact with environmental stressors, further influencing the aging process. Certain genetic variants may heighten sensitivity to environmental influences, amplifying or mitigating the impact of stressors like diet and smoking. For instance, individuals with genetic variants in the APOE gene may be more susceptible to the effects of poor diet or smoking on cognitive decline and Alzheimer’s disease risk. These gene–environment interactions, mediated by epigenetic modifications, can lead to differing aging trajectories among individuals, with some being more resilient to environmental stressors while others are more vulnerable to the development of age-related diseases [48].

### 3.1. DNA Methylation

Epigenetic clocks, utilize DNA methylation patterns to predict biological age [5]. These clocks have significant implications for identifying individuals at risk for age-related diseases and can be used to develop personalized interventions [5]. DNA methylation involves the addition of a methyl group to cytosine residues in CpG dinucleotides, leading to the formation of 5-methylcytosine (5-mC). During aging, there is a global trend of hypomethylation across various tissues, accompanied by localized hypermethylation at specific gene loci. This duality can result in the silencing of tumor suppressor genes and the activation of oncogenes, contributing to cancer development. For example, the hypermethylation of genes such as p16INK4a has been involved in the onset of cancers, including breast and colorectal cancer [49,50]. Conversely, the loss of DNA methylation can lead to genomic instability and the reactivation of transposable elements, further contributing to aging and related diseases. Age-related changes in methylation patterns are largely driven by decreased activity of DNA methyltransferases (DNMTs), particularly DNMT1, which maintains DNA methylation during cell division. This decline in DNMT activity exacerbates pro-inflammatory responses and increases the risk of chronic diseases, such as cardiovascular diseases [51], type 2 diabetes [28], various cancers [52], and neurodegenerative diseases like Alzheimer’s and Parkinson’s [53]. In addition to DNMTs, aging also leads to increased activity of ten–eleven translocation (TET) enzymes, which catalyze the removal of methyl groups from DNA. This demethylation process contributes to global hypomethylation, which has been linked to genomic instability, chromosomal aberrations, and the reactivation of transposable elements that may initiate or accelerate age-related diseases, including cancer. This balance between DNMTs and TET enzymes helps explain how methylation patterns shift with age and the subsequent impact on disease susceptibility. Furthermore, altered gene expression related to metabolism increases the susceptibility to diabetes and cardiovascular diseases, while changes in neuronal gene regulation heighten the risk of neurodegenerative disorders [54].

### 3.2. Histone Modifications

Histone modifications, such as methylation, acetylation, and phosphorylation, act as regulators of chromatin structure and gene expression. Histones are proteins around which DNA is wrapped, and modifications to these proteins can either condense the chromatin (restricting access to transcription factors) or relax it (making it accessible for transcription). In the context of aging, histone modifications have been implicated in both lifespan regulation and the onset of age-related diseases. Specifically, changes in the levels of histone marks such as H3K4me3 (an activating mark) and H3K27me3 (a repressive mark) influence chromatin compaction and gene accessibility [55]. As individuals age, there is a reduction in activating marks and an accumulation of repressive marks, which contributes to global changes in gene expression. A decrease in H3K4me3 levels has been linked to reduced transcriptional activity of genes associated with cellular repair and stress responses [56]. Conversely, the accumulation of H3K27me3 leads to a more compact chromatin structure, further restricting the accessibility of transcription factors to DNA, which can result in silencing genes crucial for cellular homeostasis [57]. Acetylation generally relaxes chromatin and promotes gene expression, but a decline in histone acetyltransferase (HAT) activity with age leads to less acetylation, contributing to age-related declines in gene expression necessary for cellular function. Age-associated hypermethylation at specific gene loci, often mediated by histone methyltransferases (HMTs), can silence tumor suppressor genes. At the same time, histone demethylation enzymes, such as TET proteins, can promote global demethylation, contributing to genomic instability and cancer progression. For example, the accumulation of repressive histone marks, like H3K9me3, can silence important genes involved in DNA repair and cell cycle regulation, thus promoting age-related diseases like cancer [4].

### 3.3. Chromatin Remodelling

Chromatin remodeling complexes are critical for modulating chromatin accessibility and regulating gene expression during aging. A significant aging-related alteration includes the loss of heterochromatin, particularly in centromeric and telomeric regions, which plays a crucial role in silencing repetitive elements and ensuring genomic stability. This heterochromatin reduction leads to the activation of retrotransposable elements, such as LINE-1, which can drive genomic instability and trigger inflammatory responses [58,59]. Furthermore, as organisms age, the diminished activity of ATP-dependent chromatin remodeling complexes, including SWI/SNF and ISWI, affects nucleosome positioning, limiting the accessibility of transcription factors to their target genes and contributing to age-related declines in cellular function [60,61]. This loss of remodeling activity impairs the transcription of genes involved in essential metabolic and stress response pathways, increasing susceptibility to metabolic disorders like insulin resistance [62].

Additionally, decreased chromatin accessibility due to remodeling dysfunction compromises DNA repair gene regulation, leading to genomic instability and a heightened risk of cancer development in aged tissues [62]. Age-related remodeling deficits have been specifically connected to neurodegenerative diseases, such as Alzheimer’s and Parkinson’s, where the disrupted chromatin structure affects the expression of genes involved in neuronal function and synaptic plasticity. These changes contribute to DNA damage accumulation and protein misfolding, which are commonly observed in neurodegenerative conditions [63,64,65,66,67].

### 3.4. Non-Coding RNAs

Non-coding RNAs (ncRNAs), including microRNAs (miRNAs) and long non-coding RNAs (lncRNAs), are crucial regulators of gene expression, significantly influencing cellular senescence and aging. Alterations in ncRNA expression profiles commonly occur with aging due to transcriptional dysregulation and environmental stressors. For instance, oxidative stress can upregulate specific miRNAs, which modulate pathways linked to inflammation and apoptosis [68]. miRNAs, small RNAs (19–22 nucleotides), primarily regulate gene expression by targeting mRNAs for degradation or translational repression [69]. This extensive regulatory network allows miRNAs to impact the biological pathways, such as cell death, proliferation, and metabolism, which are critical in aging [70]. Dysregulated miRNAs are associated with conditions like neurodegeneration, cardiovascular diseases, and cancer, where they can exacerbate disease progression [70,71].

Several miRNAs modulate aging-related pathways, including insulin/IGF-1 signaling, mTOR, and DNA damage responses, which help maintain cellular homeostasis under stress [72,73]. Some, like miR-17, have even been shown to extend lifespan in animal models by inhibiting senescence-associated pathways, highlighting their therapeutic potential for age-related conditions [74]. Although most research on miRNAs and longevity focuses on invertebrates, the underlying pathways are conserved across species, making miRNA-based therapies promising for age-related diseases [75,76,77].

LncRNAs, unlike miRNAs, are a diverse group of transcripts with tissue-specific and temporally regulated expression patterns. They regulate gene expression and chromatin structure by interacting with DNA, RNA, or proteins, often affecting transcription and chromatin remodeling. Changes in lncRNA expression have been associated with aging-related diseases, including cancer, cardiovascular disease, type II diabetes, and neurodegenerative disorders [78,79,80,81]. Targeting lncRNAs may offer a novel approach to adjust gene expression profiles linked to cellular senescence, promoting tissue repair and mitigating age-related disease progression [82].

Additionally, ncRNAs stability in blood and cerebrospinal fluid makes them promising as non-invasive biomarkers for aging. ncRNA biomarkers could facilitate early disease diagnosis and monitor therapeutic efficacy, enabling personalized medicine approaches for managing age-related conditions [83,84,85]. Expanding research into ncRNAs may not only enhance therapeutic options but also advance diagnostics and individualized treatments for aging-related diseases.

### 3.5. Epigenetic and Telomerase Clocks and Circadian Rhythm in Aging

Building upon the roles of DNA methylation, chromatin remodeling, and non-coding RNAs, epigenetic clocks—such as the Horvath clock—are valuable tools in gauging biological age through DNA methylation patterns. These clocks measure the accumulation of age-related methylation changes and help identify individuals at higher risk for age-related diseases [5]. Epigenetic clocks, which serve as dynamic indicators of biological age, offer the potential to assess anti-aging interventions and personalized therapies [8].

The telomerase clock, another aging indicator, assesses cellular aging via telomere length. Telomeres gradually shorten with each cell division due to limited telomerase activity, especially in somatic cells, leading to cellular senescence and tissue dysfunction. This attrition is particularly impactful in regenerative tissues, where cellular turnover is essential for maintaining function [86].

Circadian rhythm also significantly impacts aging by regulating metabolic, repair, and immune functions through circadian genes such as PER1 and BMAL1 [86]. Disruptions to circadian rhythm, due to lifestyle factors like irregular sleep or shift work, are associated with heightened risks of metabolic disorders, immune dysregulation, and neurodegenerative diseases [87]. Circadian disturbances induce epigenetic changes, altering gene expression and potentially accelerating biological aging [37,88].

## 4. The Future of Senolytics

Senolytics represent a therapeutic strategy designed to selectively eliminate senescent cells, which contribute to the aging process and various age-related diseases through the harmful effects of SASP. The accumulation of these cells can lead to chronic inflammation and tissue dysfunction, making their removal a promising target for improving tissue function and extending healthspan [89,90] (Figure 2).

Early studies identified senolytic agents such as dasatinib and quercetin, which have shown efficacy in preclinical models [91,92]. When used in combination, dasatinib and quercetin target multiple senescent cell types more effectively than either agent alone, with studies demonstrating the enhanced clearance of senescent cells and greater improvements in physical function in aged mice. This combination leverages dasatinib’s potency against senescent preadipocytes and endothelial cells, while quercetin effectively targets senescent human endothelial cells and fibroblasts, making them a complementary therapy with broader senolytic action and potential for enhancing tissue health and extending healthspan [89,93]. For example, Zhu et al. (2015) demonstrated that the combination of these compounds not only reduced the burden of senescent cells but also improved physical function in aged mice [93]. In recent years, the list of senolytic agents has expanded to include compounds such as fisetin [94,95,96,97] and navitoclax [98], both of which effectively induce apoptosis in senescent cells by targeting specific anti-apoptotic pathways [94]. Other agents, such as Piperlongumine and FOXO4-DRI, have shown promise in selectively eliminating senescent cells through novel mechanisms like inducing reactive oxygen species (ROS) or disrupting interactions between transcription factors and senescent cell survival pathways [99]. Studies on ABT-263 (another form of Navitoclax) have demonstrated its effectiveness in clearing senescent cells, particularly in models of hematopoietic stem cell rejuvenation [100]. Senolytics primarily function by transiently inhibiting senescent cell anti-apoptotic pathways (SCAPs), which protect these cells from apoptosis despite their damage and dysfunction. SCAPs include pathways involving BCL-2/BCL-XL, PI3K/AKT, p53/p21, and dependence receptors/tyrosine kinases. By disrupting these pathways, senolytics induce apoptosis selectively in senescent cells, which are otherwise resistant to cell death. This targeted action reduces the accumulation of senescent cells, alleviating senescence-associated pathologies and potentially improving tissue function and resilience in aged or damaged tissues [101].

The advantage of this targeted “hit-and-run” approach is that, due to the slow re-accumulation of senescent cells, intermittent dosing can be used. This regimen not only reduces the risk of side effects associated with continuous treatment but also maintains therapeutic efficacy by allowing immune cells to help clear residual senescent cells after each dose, sustaining benefits in tissue function and healthspan [97,98] (Table 1).

## 5. The Interplay Between Epigenetics and Senolytics

Emerging evidence suggests a close interaction between epigenetic modifications and cellular senescence, presenting opportunities to enhance the effectiveness of senolytic therapies. Senescent cells display distinct epigenetic alterations, including changes in DNA methylation and histone modifications, which contribute to their survival and the harmful effects of the SASP. For example, epigenetic changes like the hypermethylation of tumor suppressor genes or histone deacetylation are often associated with increased senescent cell resistance to apoptosis [102,103]. Targeting these epigenetic alterations could improve the outcomes of senolytic treatments. Recent studies have shown that inhibiting histone deacetylases (HDACs) can sensitize senescent cells to senolytic agents, enhancing their removal from tissues [104]. Additionally, epigenetic reprogramming has gained attention as a potential strategy to reverse the aging process at the cellular level [105]. Techniques like induced pluripotent stem cells (iPSCs) can reset the epigenetic markers of aging, promoting cellular rejuvenation [106]. Integrating epigenetic reprogramming with senolytic therapies offers a promising dual approach: the selective elimination of senescent cells combined with the restoration of youthful epigenetic patterns in healthy cells, leading to improved tissue function and regeneration [107].

## 6. Future Directions

As the fields of epigenetics and senolytics continue to evolve, several promising research avenues warrant further exploration. First and foremost, large-scale clinical trials are essential to evaluate the long-term safety and efficacy of senolytic therapies in humans. These studies will help establish the therapeutic potential of these agents across various age-related diseases, clarifying optimal dosing regimens and treatment durations.

A deeper understanding of the molecular mechanisms underlying the interplay between epigenetic modifications and cellular senescence is important for developing targeted interventions. Investigating how specific epigenetic changes influence the senescence process, including the SASP, can reveal new strategies for enhancing the effectiveness of senolytics. For example, research into histone modifications and DNA methylation patterns in senescent cells may identify new targets for pharmacological intervention.

Moreover, integrating omics technologies, such as genomics, transcriptomics, proteomics, and metabolomics, can elucidate the complex regulatory networks involved in aging and senescence. These interdisciplinary approaches hold promise for uncovering novel biomarkers that can predict cellular senescence and aging trajectories, as well as therapeutic targets for interventions aimed at rejuvenating aged tissues.

Additionally, exploring combination therapies that integrate senolytics with other treatments, such as epigenetic modulators or regenerative medicine techniques like induced IPCs, could enhance therapeutic outcomes. Such strategies may synergistically improve tissue regeneration and function while decreasing age-related decline.

Finally, investigating the role of lifestyle factors—such as diet, exercise, and stress management—on epigenetic modifications and cellular senescence can provide valuable insights into preventive strategies for healthy aging. Understanding how these factors influence the epigenetic landscape may lead to personalized interventions that promote longevity and healthspan.

Senolytics have shown effectiveness in preclinical studies for various conditions, including cardiovascular disease [108], kidney disease [109], diabetes [110], osteoarthritis, osteoporosis [111], hepatic [112] and pulmonary fibrosis [113], steatosis, obesity, depression, mortality related to coronavirus infections, and Alzheimer’s disease [114]. Preliminary findings indicate that the combination of dasatinib and quercetin is safe for human use and effectively reduces the burden of senescent cells [115]. In animal models, a brief intermittent treatment with senolytics has been sufficient to enhance various aspects of physical fitness, even when administered later in life. This underscores the significant potential of senotherapeutics to positively influence human health and reduce healthcare costs [116].

Despite the promising role of senescent cells as therapeutic targets, there is limited information regarding their identity and characteristics in human tissues. Understanding where and when these cells arise in humans, as well as the extent of their heterogeneity in vivo, is essential for guiding targeted therapies. Consequently, there is a pressing need to develop tools that can accurately map and identify senescent cells with both spatial and temporal precision. To address this need, the SenNet Consortium was established in 2021 with the goal of functionally characterizing the diversity of senescent cells across 18 different tissues from healthy humans throughout their lifespan at a single-cell resolution. This initiative aims to enhance our understanding of cellular senescence and its implications for aging and related diseases [117].

The implications of integrating epigenetic reprogramming with senolytic treatments are profound. Techniques such as induced iPSC technology offer exciting possibilities for rejuvenating aged cells by resetting their epigenetic marks [118]. Combining these approaches could not only enhance tissue regeneration but also potentially reverse some aspects of cellular aging [119,120].

Moreover, understanding the role of the SASP in promoting chronic inflammation underscores the importance of targeting both senescent cells and their secretory profiles to mitigate age-related diseases. The ability to selectively eliminate senescent cells while modulating their secretome could lead to significant improvements in healthspan and quality of life for aging individuals [121,122].

However, challenges remain in translating these findings into clinical practice. Issues such as identifying appropriate patient populations for treatment, determining optimal treatment protocols, and addressing potential side effects must be carefully considered in future studies. Additionally, while the promise of combining epigenetic modulation with senolytic therapy is compelling, more research is needed to understand the long-term effects and safety of such interventions. Building a robust preclinical foundation is essential for identifying effective senolytic agents, understanding optimal dosing regimens, and evaluating long-term safety and efficacy. Such foundational research will be critical to inform future clinical trials, ultimately guiding the transition from laboratory findings to practical, human-centered applications in aging and disease management.

## Figures and Tables

**Figure 1 biomolecules-15-00018-f001:**
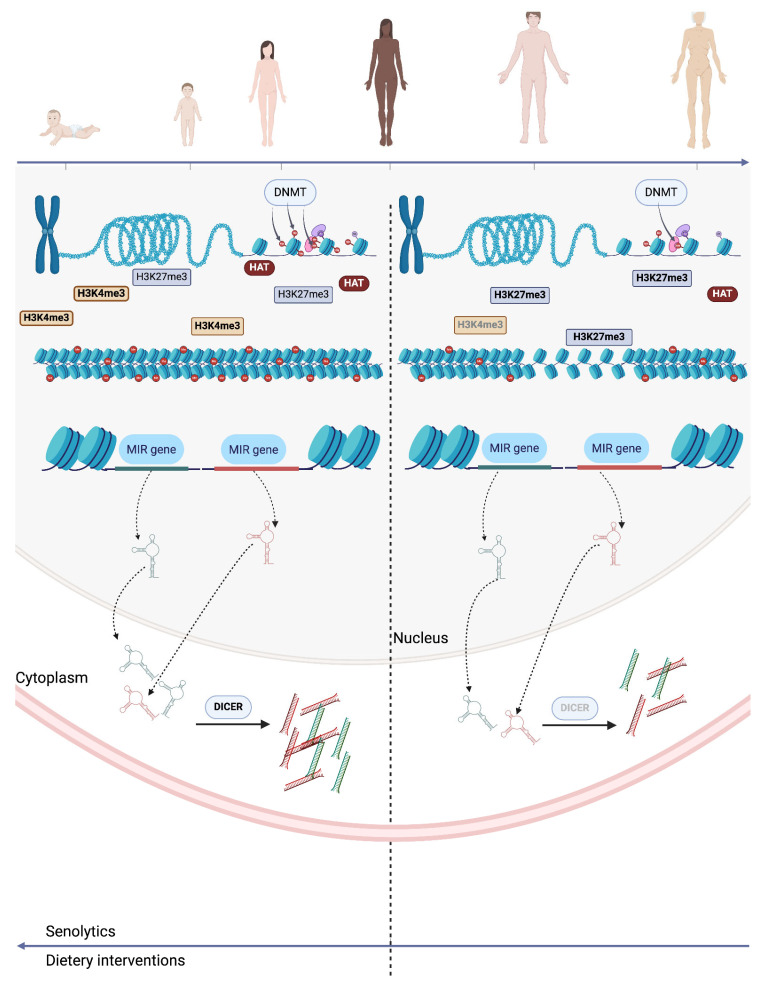
The primary epigenetic mechanisms involved in aging include DNA methylation, histone modifications, chromatin remodeling, and non-coding RNAs. During aging, there is a general trend of genome-wide hypomethylation, though specific regions may undergo either hypermethylation or hypomethylation. Aged cells also exhibit heterochromatin loss, which is reflected in changes to histone content and modification patterns. Additionally, the formation of senescence-associated heterochromatin foci (SAHFs) is a notable feature of cellular aging. Finally, miRNA deregulation, driven by impaired miRNA biogenesis, is observed across various species and tissues as part of the aging process. Abbreviations: DNMT, DNA methyltransferase; HAT, histone acetyltransferase.

**Figure 2 biomolecules-15-00018-f002:**
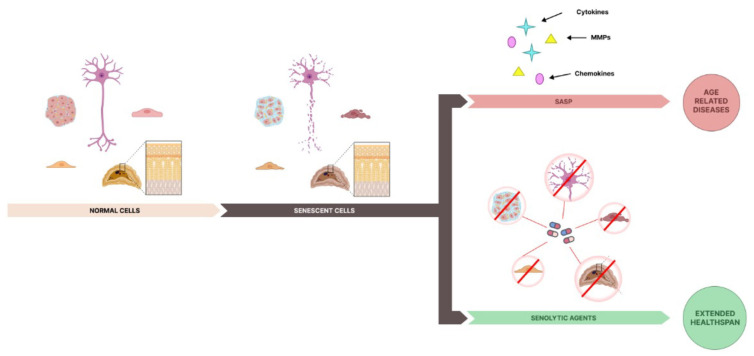
The action of senolytic agents on senescent cells. The schematic illustrates the transition from normal to senescent cells and the impact of senolytic agents on aging and healthspan. On the left, various normal cell types (e.g., neurons, fibroblasts, epithelial cells) maintain tissue function. As aging progresses, these cells accumulate damage and enter a state of senescence. Senescent cells exhibit the senescence-associated secretory phenotype (SASP), releasing inflammatory cytokines, matrix metalloproteinases (MMPs), and chemokines, which contribute to tissue dysfunction and promote age-related diseases. Senolytic agents target these dysfunctional senescent cells, as shown in the lower right section, selectively inducing apoptosis and reducing their burden. This “hit-and-run” approach helps decrease SASP factors and supports tissue homeostasis, ultimately extending healthspan by reducing inflammation and delaying the onset of age-related diseases.

**Table 1 biomolecules-15-00018-t001:** Senolytic agents and their clinical implications.

Drug	Mechanism of Action	Key Findings	Study Reference
**Dasatinib**	Pan-tyrosine kinase inhibitor	Reduces senescent cell burden; improved physical function in aged mice	[89]
**Quercetin**	Inhibits BCL-2 family proteins	Enhances apoptosis in senescent cells; improved healthspan in murine models	[91,92]
**Fisetin**	Flavonoid that targets multiple pathways	Induces apoptosis in endothelial cells; reduces markers of senescence	[94,95,96]
**Navitoclax**	BCL-2 family inhibitor	Induces apoptosis in specific senescent cell types; effective in models of osteoarthritis	[98]
**FOXO4-DRI**	Disrupts FOXO4-p53 interaction	Triggers apoptosis in senescent cells by releasing p53 into the cytosol	[99]
**Piperlongumine**	Induces reactive oxygen species (ROS) in senescent cells	Selectively induces senescent cell death via ROS-mediated pathways; reduces senescent markers in vivo.	[99]
**ABT-263**	BCL-2/BCL-xL inhibitor	Targets BCL-2/BCL-xL pathways to induce apoptosis in senescent cells; effective in reducing senescence in hematopoietic stem cells.	[100]
**D + Q (Dasatinib + Quercetin)**	Combination therapy targeting multiple senescent pathways	Improved cardiovascular function and extended healthspan in preclinical models of aged mice.	[89,93]

## Data Availability

No new data were created.

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
