# Peer review of "The Intersection of Epigenetics and Senolytics in Mechanisms of Aging and Therapeutic Approaches"

_biomolecules, 2024, doi:10.3390/biom15010018_

Round 1

Reviewer 1 Report

Comments and Suggestions for Authors

This is a review article relating to epigenetic clock and senolytics.  The manuscript comprehensively overviews the current state of knowledge regarding to these topics.  Overall, this is interesting, informative, and useful for the researchers in the aging fields.  However, there are a couple of concerns throughout the manuscript.  The authors should consider them.

1. Page 2, line 84, please specify MMPs, although it shows in the legend of Figure 2.

2. Page 3 about Figure 1, MIR gene may be microRNA gene.  Please indicate it in the legend.  As the authors mention on page 7, line 301, senolytics means to selectively eliminate senescent cells.  Senolytics in the Figure 1 seems to be unproper.  It is felt that senolytics means to rejuvenate senescent cells.  Please remove Senolytic and Dietary interventions from this Figure.

3. Page 5, line 194, please cite proper reference which shows that aging leads to increased activity of TET enzymes.

4.  Page 6, line 223, “histone demethylation enzymes, such as TET proteins,” may be wrong.

5. Page 6, line 235, are there any evidence that the activity of ATP-dependent chromatin remodeling complexes is diminished during aging?  If yes, please cite proper references.

6. Page 7, line 282, Telomere Clocks may be better than Telomerase Clocks.

7. Page 7, line 295, is citation #86 correct?

8. Page 7, line 307, may The schematic be The schematic diagram?

9. Page 8, line 313, there are two periods.

10. Page 10, line 401 and 411, are citation #116 and #117 correct?

11. Page 10, line 421, there is no reference for citation #122.

12. Page 12 in References, there is no Journal name on #30 and 38.  Indication of References is not constant.  Please recheck throughout the References.

13. Please recheck if the citations are correctly cited throughout the manuscript.

Author Response

This is a review article relating to epigenetic clock and senolytics.  The manuscript comprehensively overviews the current state of knowledge regarding to these topics.  Overall, this is interesting, informative, and useful for the researchers in the aging fields.  However, there are a couple of concerns throughout the manuscript.  The authors should consider them.

  1. Page 2, line 84, please specify MMPs, although it shows in the legend of Figure 2.

  1. Page 3 about Figure 1, MIR gene may be microRNA gene.  Please indicate it in the legend.  As the authors mention on page 7, line 301, senolytics means to selectively eliminate senescent cells.  Senolytics in the Figure 1 seems to be unproper.  It is felt that senolytics means to rejuvenate senescent cells.  Please remove Senolytic and Dietary interventions from this Figure.

  1. Page 5, line 194, please cite proper reference which shows that aging leads to increased activity of TET enzymes.

  1. Page 6, line 223, “histone demethylation enzymes, such as TET proteins,” may be wrong.

  1. Page 6, line 235, are there any evidence that the activity of ATP-dependent chromatin remodeling complexes is diminished during aging?  If yes, please cite proper references.

  1. Page 7, line 282, Telomere Clocks may be better than Telomerase Clocks.

  1. Page 7, line 295, is citation #86 correct?

  1. Page 7, line 307, may The schematic be The schematic diagram?

  1. Page 8, line 313, there are two periods.

  1. Page 10, line 401 and 411, are citation #116 and #117 correct?

  1. Page 10, line 421, there is no reference for citation #122.

  1. Page 12 in References, there is no Journal name on #30 and 38.  Indication of References is not constant.  Please recheck throughout the References.

  1. Please recheck if the citations are correctly cited throughout the manuscript.

Answer: We are thankful for the careful observation, which has all been addressed in the revised version

Reviewer 2 Report

Comments and Suggestions for Authors

This is an excellent review on epigenetics, aging and senolytics. The authors succeeded in putting together very complex layers on knowledge in a comprehensive and clear way. It is a pleasure to read and,, it may become an educational reference if properly handled.

The only suggestion would be to expand a bit on the clinical aspects of implementing senolytics therapy. Authors have briefly mentioned it in the 'Future directions " sections, but in my opinion the topic deserves further discussion. The high potential of senolytics has been widely demonstrated in preclinical models, yet very important hurdles impairs its application in patients. I would suggest discussing in more detail the following points:

- what are the main safety concerns? 

- would it be possible to target senolytic therapy to the affected organ to avoid undesired secondary effects

-  senolytics therapy is aimed to remove senescent cells but does not attack the cause of senescence. This may create a dependency of therapy for long time. What would be a plausible strategy to mitigate this?

Author Response

Reviewer #2

This is an excellent review on epigenetics, aging and senolytics. The authors succeeded in putting together very complex layers on knowledge in a comprehensive and clear way. It is a pleasure to read and,, it may become an educational reference if properly handled.

The only suggestion would be to expand a bit on the clinical aspects of implementing senolytics therapy. Authors have briefly mentioned it in the 'Future directions " sections, but in my opinion the topic deserves further discussion. The high potential of senolytics has been widely demonstrated in preclinical models, yet very important hurdles impairs its application in patients. I would suggest discussing in more detail the following points:

- what are the main safety concerns? 

- would it be possible to target senolytic therapy to the affected organ to avoid undesired secondary effects

Answer: That is a very good point, which has been addressed in the revised version

-  senolytics therapy is aimed to remove senescent cells but does not attack the cause of senescence. This may create a dependency of therapy for long time. What would be a plausible strategy to mitigate this?

Answer: Your observation is correct: senolytic therapies, while effective in removing senescent cells, do not address the root causes of cellular senescence. To mitigate the potential dependency on long-term therapy, a comprehensive strategy could involve addressing both the removal of senescent cells and the underlying drivers of senescence.

Reviewer 3 Report

Comments and Suggestions for Authors

Dear authors,

The present article entitled “The Intersection of Epigenetics and Senolytics in Mechanisms of Aging and Therapeutic Approaches” by Burdusel et al. describes on the one hand, how dysfunctional epigenetic mechanisms can drive cells into senescence and on the other hand mentions existing experimental treatments targeting the removal of senescence cells. In figure 1 the authors suggest that age-related epigenetic changes could be reverted by dietary interventions and/or senolytics (at least this is what one would interpret from the arrow displayed at the bottom of the figure). 

The article is well-written and address an interesting and, as the authors mention, a wide unexplored topic. However, I have some concerns: i) the structure and narrative fits rather to a book chapter, ii) the side effects of (some) senolytics are not mentioned, iii) the authors propose to explore therapy strategies targeting epigenetic changes and apply these additionally to senolytics. However, in case that epigenetic strategies could reverse aging and promote rejuvenation, the use of senolytics would not be necessary anymore (?)

Recommended changes:

i)               Paragraph 3. Epigenetic of Aging describes life style factors that might disturb epigenetic mechanisms leading to accelerated aging (or cellular senescence). This section should be called: “Environmental factors influencing epigenetic mechanisms might accelerate ageing processes” (or similar). A separate section should be dedicated to the mechanisms such as DNA methylation, histone modifications, chromatin remodelling, and non-coding 107 RNAs. Epigenetic and Telomerase Clocks, and Circadian Rhythm section needs to be assigned to the right section: Sleep quality and quantity is a life style, while cellular circadian rhythm pathways belongs to mechanisms section. 

ii)              Table 1 should include known and potential side effects as well as those senolytics that actually made it into clinical trials and those which are still experimental and has not been tested in humans yet. 

iii)            It is not clear to me how a combined therapy would look like and what would be the consequences for the young healthy cells and if rejuvenation is achieved, to what extend are senolytics still needed and why? This needs to be discussed.

General major comments

After making the point that environmental factors induce epigenetic changes which might accelerate ageing processes, the authors propose to explore a “new group of senolytics” which could protect cells from those changes. 

In section 5, (the central part of the manuscript but the shortest) the authors mention two examples/directions:

1.     “Recent studies have shown that inhibiting histone deacetylases (HDACs) can sensitize senescent cells to senolytic agents, enhancing their removal from tissues [104].”

Indeed, this would be in addition to senolytics aiming a better outcome

2.     “… epigenetic reprogramming has gained attention as a potential strategy to reverse the aging process at the cellular level”

In this case, and based on the definition provided above (Senolytics represent a therapeutic strategy designed to selectively eliminate senescent cells) senolytics would be replaced by reprograming strategies since this strategy reverts or prevents senescence rather than eliminate senescence cells.

Maybe the title “Environmental factors-induced epigenetic changes as a new target for ageing therapy” or “Environmental factors-induced epigenetic changes as a new target for removing/reprograming senescent cells” describes better the contents (?).

This work doesn´t provide new information. However, it reviews the literature on environmental factors inducing epigenetic alterations which might drive cells into senescence. Based on this comprehensive summary the authors recognize a gap in the field of senolytics research and suggest new research directions. This idea might be appreciated by scientists and is worth of publication. However, the recommendations mentioned above should be addressed before I agree with publication.

Minor comments

Line 73: please replace “stable” through “irreversible”

Figure 1: explain arrow

Line 294: please replace “aging” through “aging process”

Line 313: delete one dot

Author Response

Reviewer #3

The present article entitled “The Intersection of Epigenetics and Senolytics in Mechanisms of Aging and Therapeutic Approaches” by Burdusel et al. describes on the one hand, how dysfunctional epigenetic mechanisms can drive cells into senescence and on the other hand mentions existing experimental treatments targeting the removal of senescence cells. In figure 1 the authors suggest that age-related epigenetic changes could be reverted by dietary interventions and/or senolytics (at least this is what one would interpret from the arrow displayed at the bottom of the figure).

Answer: Thank you for the observation. Dietary restriction may help delay aging, but in this context, it has been removed from the figure.

The article is well-written and address an interesting and, as the authors mention, a wide unexplored topic. However, I have some concerns: i) the structure and narrative fits rather to a book chapter, ii) the side effects of (some) senolytics are not mentioned, iii) the authors propose to explore therapy strategies targeting epigenetic changes and apply these additionally to senolytics. However, in case that epigenetic strategies could reverse aging and promote rejuvenation, the use of senolytics would not be necessary anymore (?)

Answers. In the revised version, the side effects of senolytics have been mentioned (lines 387-390; 418-430)

Recommended changes:

Paragraph 3. Epigenetic of Aging describes life style factors that might disturb epigenetic mechanisms leading to accelerated aging (or cellular senescence). This section should be called: “Environmental factors influencing epigenetic mechanisms might accelerate ageing processes” (or similar).

Answer: The title of the section has been changed to: Lifestyle and environmental factors influencing epigenetic mechanisms may accelerate the aging process

A separate section should be dedicated to the mechanisms such as DNA methylation, histone modifications, chromatin remodelling, and non-coding 107 RNAs.

Answer: Done

Epigenetic and Telomerase Clocks, and Circadian Rhythm section needs to be assigned to the right section

Answer: We created a new section with the title: The Impact of the Circadian Clock on Age-Associated Genetic and Epigenetic Changes

The Impact of the Circadian Clock on Age-Associated Genetic and Epigenetic Changes

Sleep quality and quantity is a life style, while cellular circadian rhythm pathways belongs to mechanisms section.

Answer: Some of us are psychiatrists and do not agree with dissociating quality from day/night circadian rhythmicity

Table 1 should include known and potential side effects as well as those senolytics that actually made it into clinical trials and those which are still experimental and has not been tested in humans yet. 

Answer: Done (see, Table 1)

It is not clear to me how a combined therapy would look like and what would be the consequences for the young healthy cells and if rejuvenation is achieved, to what extend are senolytics still needed and why? This needs to be discussed.

Answer: There are several major safety concerns regarding the use of senolytics to delay the aging process. Senolytics might unintentionally harm non-senescent cells, especially proliferative cells like stem cells or immune cells, leading to unintended tissue damage. The safety concerns over the use of senolytic drugs in human trials can be summarized as follows: Senescence is a natural tumor-suppressing mechanism that halts the proliferation of damaged cells. Removing senescent cells may inadvertently increase the risk of tumorigenesis, especially if residual DNA damage or mutations remain unchecked. Another concern is associated with toxicity. For example, Dasatinib and Quercetin are known to have side effects such as gastrointestinal disturbances, hepatotoxicity, and cytopenias. Navitoclax (a BCL-2 inhibitor) is associated with thrombocytopenia and neutropenia, posing a significant risk to older populations who are already prone to these conditions. Similarly, senolytics induce apoptosis in senescent cells, potentially causing acute inflammatory responses, especially in individuals with compromised immune systems or chronic inflammation [Healey et al., 2024; Riessland & Orr, 2023]-(lines lines 387-390; 418-430).

Answer: Aging affects cells and tissues in the human body in many different ways, both in terms of time and specificity. Given the hierarchical organization and integrative physiology of our body, it is very unlikely that a single therapy will ever succeed in delaying the aging process. Therefore, those working on anti-aging therapies will always consider exploring combination therapies that integrate senolytics with other treatments—too many to mention here. In the manuscript, we provided just a few examples, but there are certainly many other options, depending on the experience of the researchers involved. Our challenge is that we can test any combination in animal models, but we must be aware that they may not succeed in humans. Dietary restriction is a very good example.

General major comments

After making the point that environmental factors induce epigenetic changes which might accelerate ageing processes, the authors propose to explore a “new group of senolytics” which could protect cells from those changes. 

 Answer: Unfortunately, we could not locate the “new group of senolytics” and appreciate getting more information on this.

In section 5, (the central part of the manuscript but the shortest) the authors mention two examples/directions:

  1. “Recent studies have shown that inhibiting histone deacetylases (HDACs) can sensitize senescent cells to senolytic agents, enhancing their removal from tissues [104].”

 Indeed, this would be in addition to senolytics aiming a better outcome

Answer: We agree

  1. “… epigenetic reprogramming has gained attention as a potential strategy to reverse the aging process at the cellular level”

Answer: In this case, and based on the definition provided above (Senolytics represent a therapeutic strategy designed to selectively eliminate senescent cells) senolytics would be replaced by reprograming strategies since this strategy reverts or prevents senescence rather than eliminate senescence cells.

Maybe the title “Environmental factors-induced epigenetic changes as a new target for ageing therapy” or “Environmental factors-induced epigenetic changes as a new target for removing/reprograming senescent cells” describes better the contents (?).

 Answer: The suggestions are interesting. However, we prefer to keep the original title. We mentioned 'reprogramming strategies' because we work on this and believe that medical genetics is the future of medicine, also for aging research.

This work doesn´t provide new information. However, it reviews the literature on environmental factors inducing epigenetic alterations which might drive cells into senescence. Based on this comprehensive summary the authors recognize a gap in the field of senolytics research and suggest new research directions. This idea might be appreciated by scientists and is worth of publication. However, the recommendations mentioned above should be addressed before I agree with publication.

Minor comments

Line 73: please replace “stable” through “irreversible”

Figure 1: explain arrow

Line 294: please replace “aging” through “aging process”

Line 313: delete one dot

 Answer: Thank you for your careful reading. We corrected the text as suggested.
